# The awareness of public about the Emergency Medical Services in the Eastern region of Saudi Arabia

Ahmed Alanazy[1,2,3]*, Abdullah Alruwaili[1,2,3], Saleh Alswaidan[1,2,3], Hassan Alobaid[1,2,3], Ahmed Alomran[1,2,3], Abdulkarim Hzazi[1,2,3], Ibrahim Alhussain[1,2,3], Meshal Alharbi[4,5], Meshary Binhotan[4,5]

1 Emergency Medical Services Department, College of Applied Medical Sciences, King Saud Bin Abdulaziz University for Health Sciences, Al Ahsa, Saudi Arabia, 2 King Abdullah International Medical Research Center, Al Ahsa, Saudi Arabia, 3 Ministry of National Guard—Health Affairs, Al Ahsa, Saudi Arabia, 4 Emergency Medical Services Department, College of Applied Medical Sciences, King Saud Bin Abdulaziz University for Health Sciences, Riyadh, Saudi Arabia, 5 King Abdullah International Medical Research Center, Riyadh, Saudi Arabia

* alanazya@ksau-hs.edu.sa

**Data Availability Statement:** All data files are available from the Harvard Dataverse database DOI: https://doi.org/10.7910/DVN/BPVQET.

## Abstract

Emergency Medical Services (EMS) are crucial for immediate medical assistance during life-threatening situations. However, insufficient public awareness about EMS services can impede their effectiveness. This study aimed to assess EMS knowledge and trust among the population of Eastern Saudi Arabia while identifying factors contributing to low awareness. A cross-sectional study was conducted in Eastern Saudi Arabia from September 2022 to September 2023. The study included participants aged 18 to 60 from diverse backgrounds. Using a convenience sampling approach, data was collected using a validated questionnaire covering demographics, hypothetical scenarios, EMS knowledge, and trust in EMS. We conducted the Chi-square tests and logistic regression using Jamovi software, with significance levels set at p < 0.05. Our study yielded 435 participants; 55% were males. Gender-based analysis showed significant differences in responses regarding first aid provision and EMS services (P < 0.001). Expectations for EMS response times also varied by gender (P = 0.01). Knowledge-based analysis revealed that age and education significantly influenced EMS knowledge (P < 0.001). Respondents with EMS knowledge were more likely to know how to provide first aid, understand the importance of emergency number 112, and trust EMS (P < 0.001). Trust-based analysis showed age and education-related differences in EMS trust (P < 0.001). Respondents with EMS knowledge and awareness of emergency numbers displayed higher trust in EMS (P < 0.001). This study underscores the need for enhanced public awareness of EMS services in Eastern Saudi Arabia. Age, education, and gender emerged as critical factors affecting EMS knowledge and trust. Bridging this awareness gap necessitates tailored educational campaigns and continuous monitoring. Policymakers should prioritise EMS awareness within broader healthcare strategies, contributing to improved public health outcomes and community well-being.

**Funding:** The author(s) received no specific funding for this work.

**Competing interests:** The authors have declared that no competing interests exist.

# 1. Introduction

Introduction Emergency Medical Services (EMS) constitute a vital component within the global healthcare system, functioning as the primary responders to critical medical incidents and accidents [1]. EMS encompasses services and resources meticulously designed to provide immediate medical attention, transportation, and aid to individuals facing life-threatening emergencies [2]. These services fulfil an indispensable role by delivering prompt and expert patient care, consequently mitigating the morbidity and mortality rates associated with medical conditions that necessitate rapid intervention [3, 4].

EMS serves the foundational purpose of bridging the gap between the emergence of an emergency and the inception of medical treatment, ensuring that individuals receive timely and life-preserving interventions. Effective EMS systems are critical in preserving public health, offering invaluable pre-hospital care [4, 5].

Prior research has shed light on the significance of public awareness concerning EMS and its implications for emergency response effectiveness. Studies conducted in Saudi Arabia and other developing countries have explored the extent to which the general population is informed about EMS services and their capabilities [6–8]. These investigations have consistently revealed a noticeable gap in public knowledge regarding EMS functions, availability, and how to access these critical services. Despite the essential role of EMS in saving lives, many individuals lack basic awareness of its existence, hindering the system's ability to provide timely assistance during emergencies.

The inadequacy in public awareness of Emergency Medical Services (EMS) is a pressing concern that extends beyond regional confines, encompassing not only Eastern Saudi Arabia but potentially the entirety of the Kingdom of Saudi Arabia and beyond. This deficiency poses significant implications for public health and safety on a broader scale. The failure of individuals to promptly identify emergency signs, access EMS services expediently, and provide crucial information to emergency dispatchers jeopardizes timely care delivery and can compromise health outcomes not just locally but potentially on a global level [9, 10].

This cross-sectional study aims to assess the extent of knowledge regarding EMS within the Eastern Region of Saudi Arabia and the level of trust the local population places in EMS. Additionally, the research aims to pinpoint the underlying factors contributing to the prevalent lack of awareness about EMS in this region.

# 2. Methods

This cross-sectional study was conducted in the Eastern region of Saudi Arabia from September 2022 to September 2023.

## 2.1. Inclusion criteria

We included participants of both genders, aged between 18 and 60 years. Exclusion criteria included individuals associated with health professions.

## 2.2. Sample size

We determined the sample size using the online tool available at https://www.qualtrics.com/blog/calculating-sample-size/. Employing a confidence level of 95%, an estimated population size of 5,028,735, and a margin of error of 5%, the ideal sample size was calculated as 385 participants (193 males and 192 females). Hence, our study aimed to include 385 participants from Eastern Saudi Arabia.

## 2.3. Questionnaire

Our survey consisted of 22 questions. Demographic questions included age, gender, and education level. They were also asked about hypothetical scenarios, such as what actions they would take if they encountered an unconscsious person by the roadside. Knowledge of EMS, awareness of the unified emergency number 112, and understanding of the services provided by EMS were also assessed. Additionally, participants were questioned about their opinions on paramedics treating patients in the field, the adequacy of EMS coverage in the Eastern region, and the necessity of MEDEVAC services. Trust in EMS, attitudes toward paramedics entering homes without male escorts, and the acceptance of paramedics refusing non-urgent transport were explored. Response times for EMS and behavior when encountering an ambulance on the road were also investigated. The survey delved into participants' familiarity with Tawakkalna (the official Saudi contact tracing app endorsed by the Ministry of Health) for ambulance calls. Appendix 1

## 2.4. Data collection methods, instrument used, measurements

The online survey, created in Arabic using Google Forms, employed a convenience sampling technique by initially distributing the survey link via various social media channels like LinkedIn, WhatsApp, and Telegram. Additionally, participants were encouraged to share the survey link within their networks, thereby expanding the sample through referrals from existing respondents. Incorporating a targeted approach, the survey began with a screening question asking respondents if they resided in the Eastern region. Responding "Yes" granted access to the full survey, ensuring focused data collection from the intended demographic. This strategy effectively balanced the need for specific geographic participation with the inclusion of individuals from other regions in subsequent survey sections.To validate the questionnaire, a pilot study was conducted. Feedback from five experts in the field was obtained before commencing the actual data collection process. We conducted a reliability analysis using Cohen's kappa test (with a value of 0.773), indicating a positive level of internal consistency in questionnaire scales.

## 2.5. Data management and analysis plan

The collected data were analyzed using the Jamovi software based on R programming. The Chi-square test was utilized to analyze categorical variables presented as frequencies. A significance level of $p < 0.05$ was used as the threshold for determining statistical significance. We analyzed the data based on gender, EMS knowledge, and EMS trust. Also, we conducted an adjusted and non-adjusted logistic regression analysis for demographics affecting EMS knowledge. We presented the results using tables and figures to improve clarity and understanding.

## 2.6. Ethical considerations

Ethical approval was obtained from the king Abdullah international medical research centerre view board before the study commenced. The approval number is IRB/1868/22. Every participant's informed consent was diligently acquired before their involvement in the study. In the online survey, informed consent was obtained by presenting a detailed information sheet at the beginning of the questionnaire. Participants were provided with comprehensive information about the study's purpose, procedures, potential risks, confidentiality measures, and their rights as participants. Prior to proceeding with the survey, participants were required to actively indicate their consent by acknowledging their understanding of the provided

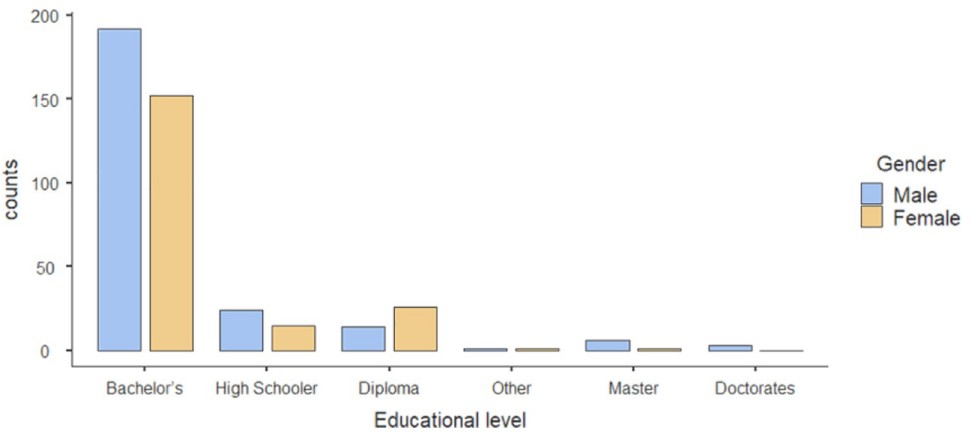

**Fig 1. Educational level according to gender distribution.**

information and agreeing to participate. Throughout the research, we preserved the confidentiality and anonymity of all participants' data.

## 3. Results

Our study included 435 participants, and 240 (55%) were males. Most participants were 18–24 years, 190 (43.7%), and 25–34 years 142 (32.6%). Regarding educational level, the majority of participants had a bachelor's degree (344 (79.1%), and a smaller percentage were high schoolers 39 (9%), and 40 (9.2%) had a diploma. **Figs 1 and 2**.

### 3.1. Gender-based analysis

When providing first aid to an unconscious person, a significant gender difference ($P < 0.001$) was observed, with more males 103 (43%) choosing to call the emergency response service, while only 42 (22%) of females chose to call the emergency response services. Also, A significant difference was observed regarding services offered by EMS ($P = 0.004$), 123 (51%) of

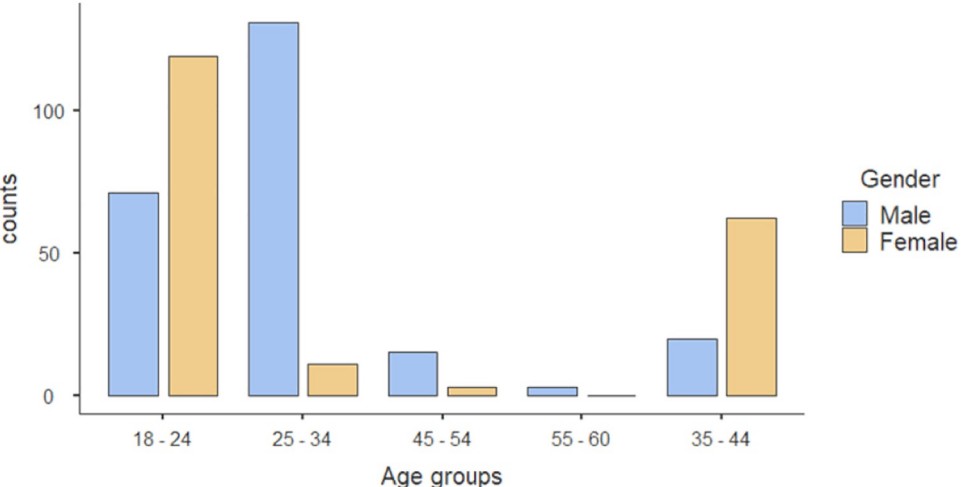

**Fig 2. Age groups according to gender distrubution.**

males chose both transferring and treating the patient answer, while 122 (63%) of females chose only patient transfer.

Expectations for EMS response times to arrive at their homes showed a significant difference (P = 0.01), with 143 (60%) of males and 109 (56%) of females anticipating a 30-minute response. In comparison, 32 (13%) males and 48 (25%) females expected a quicker 10-minute arrival.

The study also revealed a significant difference in the choice of paramedics to decline non-urgent transport (<0.001). More males, 79 (33%), chose to decline non-urgent transport than females, 33 (17%).

Furthermore, a significant difference existed in the response to an ambulance behind on the road (P = 0.001). A larger proportion of 117 (49%) males and 125 (64%) females moved out of the way only if the sirens, lights, and radio communication (SRC) were behind. Moreover, the rating of current EMS quality showed a significant difference (P = 0.017), as most males 112 (47%) and females 96 (49%) rated it as poor.

In contrast, several variables, such as EMS knowledge, emergency number 112 awareness, EMS contact number, trust of EMS, and Tawakkalna knowledge, did not show significant gender differences. Further details are shown in Table 1.

## 3.2. EMS knowledge-based analysis

Regarding age groups, a significant difference emerged (p < 0.001). The 18–24 age group exhibited the highest knowledge levels, with 132 (76%) responding affirmatively.

The educational level also significantly affected EMS knowledge (p < 0.001). The knowledge levels of individuals with a Bachelor's degree were significantly lower, as they comprised 244 (93%) of all negative knowledge.

Regarding specific emergency scenarios, respondents who possessed EMS knowledge (p < 0.001) were more likely to know how to provide first aid to an unconscious person, call emergency response services, and understand the importance of Emergency Number 112.

Furthermore, respondents with EMS knowledge (p < 0.001) were more familiar with the contact numbers for EMS services, different services offered by EMS, and the role of a dispatcher in the EMS system.

Regarding response times, knowledge-based respondents (p < 0.001) had a better understanding of EMS response times and appropriate actions to take when encountering an ambulance on the road.

Lastly, knowledge-based respondents (p < 0.001) expressed a higher level of trust in EMS, were more likely to allow paramedics' entry into their homes in emergencies without male presence, and recognized the need for MEDEVAC services. Table 2.

## 3.3. EMS trust-based analysis

Age groups significantly impacted EMS trustworthiness (P < 0.001). Notably, the 18–24 age group exhibited the highest trust levels at 149 (66%).

The educational level also played a vital role in trust (P < 0.001). The trust levels of respondents with a Bachelor's degree were lower as they made up 208 (99%) of all the negative trust reported.

Individuals with EMS knowledge (P < 0.001) displayed higher trust in EMS services regarding specific emergency scenarios. Respondents who were aware of Emergency Number 112 (P < 0.001) and the contact number for EMS services (P <0.001) demonstrated higher trust in EMS.

Table 1. Gender-based analysis. Analysis was conducted using Pearson's Chi-squared test.

| Characteristic (Total = 435) | Male N = 240 (55%) | Female N = 195 (45%) | p-value |
|---|---|---|---|
| **Provide First Aid to unconscious Person** | | | <0.001 |
| Call the emergency response services | 103 (43%) | 42 (22%) | |
| Call the people standing nearby | 6 (2.5%) | 29 (15%) | |
| Try to awaken the person | 18 (7.5%) | 25 (13%) | |
| Shift to the nearby hospital | 2 (0.8%) | 4 (2.1%) | |
| Ignore and walk away | 1 (0.4%) | 0 (0%) | |
| Call the police | 110 (46%) | 95 (49%) | |
| Emergency Number 112 Awareness | 186 (78%) | 158 (81%) | 0.37 |
| **EMS Contact number** | | | 0.093 |
| 997 | 105 (44%) | 88 (45%) | |
| 998 | 123 (51%) | 105 (54%) | |
| 996 | 6 (2.5%) | 0 (0%) | |
| 999 | 6 (2.5%) | 2 (1.0%) | |
| **Services Offered by EMS** | | | 0.004 |
| Transferring and treating the patient | 123 (51%) | 73 (37%) | |
| Patient transfer only | 117 (49%) | 122 (63%) | |
| **Role of a Dispatcher** | | | 0.56 |
| Receive call only | 136 (57%) | 105 (54%) | |
| Receiving calls and providing medical advice | 104 (43%) | 90 (46%) | |
| **EMS Response Time to arrive home** | | | 0.01 |
| 10 minutes | 32 (13%) | 48 (25%) | |
| 30 minutes | 143 (60%) | 109 (56%) | |
| 15 minutes | 63 (26%) | 36 (18%) | |
| 1 hour | 2 (0.8%) | 2 (1.0%) | |
| **Response to ambulance behind on Road** | | | 0.001 |
| Immediately get out of the way | 75 (31%) | 32 (16%) | |
| Move out of the way only if lights are on | 47 (20%) | 37 (19%) | |
| I don't move | 1 (0.4%) | 1 (0.5%) | |
| Move only if SRC is behind me | 117 (49%) | 125 (64%) | |
| **Actions when encountering EMS or EMT** | | | 0.62 |
| Stay away | 70 (29%) | 46 (24%) | |
| Interfering and helping them with permission | 57 (24%) | 49 (25%) | |
| Just watching | 112 (47%) | 99 (51%) | |
| Interfering and helping them without permission | 1 (0.4%) | 1 (0.5%) | |
| Field Treatment by Paramedics | 70 (29%) | 88 (45%) | <0.001 |
| Paramedics' choice to decline non-urgent transport | 79 (33%) | 33 (17%) | <0.001 |
| **Overall Quality of Current EMS Service** | | | 0.017 |
| Good | 48 (20%) | 57 (29%) | |
| Excellent | 65 (27%) | 37 (19%) | |
| Average | 15 (6.2%) | 5 (2.6%) | |
| Poor | 112 (47%) | 96 (49%) | |

In terms of response times, respondents who trusted EMS (P < 0.001) had a better understanding of EMS response times and appropriate actions to take when encountering an ambulance on the road.

**Table 2. EMS trust-based analysis and trust-based analysis.** Analysis was conducted using Pearson's Chi-squared test.

| Variables | EMS knowledge | | EMS trust | | |
|---|---|---|---|---|---|
| | Yes N = 173 (40%) | No N = 262 (60%) | Yes N = 225 (52%) | No N = 210 (48%) | P-value |
| **Age groups** | | | | | **<0.001** |
| 18–24 | 132 (76%) | 58 (22%) | 149 (66%) | 41 (20%) | |
| 25–34 | 17 (9.8%) | 125 (48%) | 31 (14%) | 111 (53%) | |
| 35–44 | 15 (8.7%) | 67 (26%) | 25 (11%) | 57 (27%) | |
| 45–54 | 8 (4.6%) | 10 (3.8%) | 17 (7.6%) | 1 (0.5%) | |
| 55–60 | 1 (0.6%) | 2 (0.8%) | 3 (1.3%) | 0 (0%) | |
| **Educational level** | | | | | **<0.001** |
| Bachelor's | 100 (58%) | 244 (93%) | 136 (60%) | 208 (99%) | |
| High Schooler | 24 (14%) | 15 (5.7%) | 38 (17%) | 1 (0.5%) | |
| Diploma | 40 (23%) | 0 (0%) | 40 (18%) | 0 (0%) | |
| Master | 4 (2.3%) | 3 (1.1%) | 7 (3.1%) | 0 (0%) | |
| Doctorates | 3 (1.7%) | 0 (0%) | 2 (0.9%) | 1 (0.5%) | |
| Other | 2 (1.2%) | 0 (0%) | 2 (0.9%) | 0 (0%) | |
| **Provide First Aid to unconscious Person** | | | | | **<0.001** |
| Call the emergency response services | 104 (60%) | 41 (16%) | 141 (63%) | 4 (1.9%) | |
| Call the people standing nearby | 32 (18%) | 3 (1.1%) | 35 (16%) | 0 (0%) | |
| Try to awaken the person | 33 (19%) | 10 (3.8%) | 43 (19%) | 0 (0%) | |
| Shift to the nearby hospital | 4 (2.3%) | 2 (0.8%) | 5 (2.2%) | 1 (0.5%) | |
| Ignore and walk away | 0 (0%) | 1 (0.4%) | 1 (0.4%) | 0 (0%) | |
| Call the police | 0 (0%) | 205 (78%) | 0 (0%) | 205 (98%) | |
| **Emergency Number 112 awareness** | 92 (53%) | 252 (96%) | 136 (60%) | 208 (99%) | **<0.001** |
| **EMS Contact number** | | | | | **<0.001** |
| 997 | 154 (89%) | 39 (15%) | 190 (84%) | 3 (1.4%) | |
| 998 | 12 (6.9%) | 216 (82%) | 23 (10%) | 205 (98%) | |
| 996 | 1 (0.6%) | 5 (1.9%) | 5 (2.2%) | 1 (0.5%) | |
| 999 | 6 (3.5%) | 2 (0.8%) | 7 (3.1%) | 1 (0.5%) | |
| **Services Offered by EMS** | | | | | **<0.001** |
| Transferring and treating the patient | 143 (83%) | 53 (20%) | 191 (85%) | 5 (2.4%) | |
| Patient transfer only | 30 (17%) | 209 (80%) | 34 (15%) | 205 (98%) | |
| **Role of a Dispatcher** | | | | | **<0.001** |
| Receive call only | 26 (15%) | 215 (82%) | 35 (16%) | 206 (98%) | |
| Receiving calls and providing medical advice | 147 (85%) | 47 (18%) | 190 (84%) | 4 (1.9%) | |
| **EMS Response Time to arrive home** | | | | | **<0.001** |
| 10 minutes | 61 (35%) | 19 (7.3%) | 79 (35%) | 1 (0.5%) | |
| 30 minutes | 28 (16%) | 224 (85%) | 46 (20%) | 206 (98%) | |
| 15 minutes | 81 (47%) | 18 (6.9%) | 97 (43%) | 2 (1.0%) | |
| 1 hour | 3 (1.7%) | 1 (0.4%) | 3 (1.3%) | 1 (0.5%) | |
| **Response to ambulance behind on Road** | | | | | **<0.001** |
| Immediately get out of the way | 75 (43%) | 32 (12%) | 105 (47%) | 2 (1.0%) | |
| Move out of the way only if lights are on | 62 (36%) | 22 (8.4%) | 81 (36%) | 3 (1.4%) | |
| I don't move | 2 (1.2%) | 0 (0%) | 2 (0.9%) | 0 (0%) | |
| Move only if SRC is behind me | 34 (20%) | 208 (79%) | 37 (16%) | 205 (98%) | |
| **Actions when encountering EMS or EMT** | | | | | **<0.001** |
| Stay away | 73 (42%) | 43 (16%) | 113 (50%) | 3 (1.4%) | |
| Interfering and helping them with permission | 93 (54%) | 13 (5.0%) | 105 (47%) | 1 (0.5%) | |
| Just watching | 6 (3.5%) | 205 (78%) | 5 (2.2%) | 206 (98%) | |
| Interfering and helping them without permission | 1 (0.6%) | 1 (0.4%) | 2 (0.9%) | 0 (0%) | |

*(Continued)*

**Table 2.** (Continued)

| Variables | EMS knowledge | | EMS trust | | |
|---|---|---|---|---|---|
| | **Yes N = 173 (40%)** | **No N = 262 (60%)** | **Yes N = 225 (52%)** | **No N = 210 (48%)** | **P-value** |
| **Field Treatment by Paramedics** | 122 (71%) | 36 (14%) | 153 (68%) | 5 (2.4%) | **<0.001** |
| **Enough EMS Coverage in east** | 118 (68%) | 38 (15%) | 155 (69%) | 1 (0.5%) | **<0.001** |
| **Need of MEDEVAC Services** | 165 (95%) | 52 (20%) | 213 (95%) | 4 (1.9%) | **<0.001** |
| **Paramedics' entry into home in emergency Without Male presence** | 154 (89%) | 45 (17%) | 196 (87%) | 3 (1.4%) | **<0.001** |
| **Paramedics' choice to decline non-urgent transport** | 91 (53%) | 21 (8.0%) | 109 (48%) | 3 (1.4%) | **<0.001** |
| **Tawakkalna knowledge** | 102 (59%) | 15 (5.7%) | 115 (51%) | 2 (1.0%) | **<0.001** |
| **Overall Quality of Current EMS Service** <br> **<0.001** | | | | | **<0.001** |
| **Good** | 84 (49%) | 21 (8.0%) | 104 (46%) | 1 (0.5%) | |
| **Excellent** | 73 (42%) | 29 (11%) | 101 (45%) | 1 (0.5%) | |
| **Average** | 13 (7.5%) | 7 (2.7%) | 18 (8.0%) | 2 (1.0%) | |
| **Poor** | 3 (1.7%) | 205 (78%) | 2 (0.9%) | 206 (98%) | |

Furthermore, trust-based respondents (P < 0.001) expressed a higher likelihood of allowing paramedics' entry into their homes in emergencies without male presence and recognized the need for MEDEVAC services. Table 2.

## 3.2. Logistic regression analysis for baseline factors affecting EMS knowledge

In the non-adjusted model, the intercept value suggests that the baseline odds of having EMS knowledge were 0.681, indicating a moderate likelihood of possessing such knowledge. Gender differences, as indicated by the "Male–Female" estimate, were not statistically significant (p = 0.775), meaning that there was no significant gender-based difference in EMS knowledge. Among age groups, only individuals in the "18–24–55–60" category showed a significantly higher likelihood of having EMS knowledge (p = 0.22).

Educational levels played a significant role, with "Bachelor's–High Schooler" showing a substantial negative effect on EMS knowledge (p < 0.001), indicating that those with a Bachelor's degree were less likely to have EMS knowledge. However, the "Diploma–High Schooler" and "Doctorates–High Schooler" estimates were extraordinarily high, but their p-values were insignificant, suggesting these categories did not provide meaningful insights into EMS knowledge.

In the adjusted model, gender became a significant predictor (p < 0.001), with males being nearly four times more likely to possess EMS knowledge than females. The "18–24–55–60" age group showed a significantly higher likelihood of having EMS knowledge (p = 0.004). Educational levels also influenced EMS knowledge, with "Bachelor's–High Schooler" showing a negative effect (p = 0.001), indicating that having a Bachelor's degree was associated with lower EMS knowledge. Table 3.

## 4. Discussion

Our study revealed that men were more likely to call EMS when they encountered an unconscious person and yielded to ambulances with lights and sirens. This suggests a need for gender-specific awareness campaigns [11–13]. While males demonstrated greater awareness in certain aspects, females, on the other hand, were more inclined to view patient transfer as the primary function of EMS. This underscores the importance of addressing misconceptions about the roles and capabilities of EMS services.

**Table 3. Adjusted and non-adjusted logistic regression analysis.**

**Model Coefficients—EMS knowledge**

| Non-adjusted model | | | | |
|---|---|---|---|---|
| | | | **95% Confidence Interval** | |
| **Predictor** | **p** | **Odds ratio** | **Lower** | **Upper** |
| Intercept | 0.008 | 0.681 | 0.512 | 0.906 |
| Gender: | | | | |
| Male–Female | 0.775 | 0.945 | 0.643 | 1.391 |
| Intercept | 0.571 | 0.5 | 0.0453 | 5.51 |
| Age groups: | | | | |
| 18–24–55–60 | 0.22 | 4.552 | 0.4047 | 51.2 |
| 25–34–55–60 | 0.298 | 0.272 | 0.0234 | 3.16 |
| 45–54–55–60 | 0.72 | 1.6 | 0.1219 | 20.99 |
| 35–44–55–60 | 0.523 | 0.448 | 0.0381 | 5.27 |
| Intercept | 0.153 | 1.6 | 0.839 | 3.05 |
| Educational level: | | | | |
| Bachelor's–High Schooler | < .001 | 0.256 | 0.129 | 0.509 |
| Diploma–High Schooler | 0.978 | 2.66E+07 | 0 | Inf |
| Master–High Schooler | 0.826 | 0.833 | 0.163 | 4.253 |
| Doctorates–High Schooler | 0.994 | 2.66E+07 | 0 | Inf |
| Adjusted model | | | | |
| | | | **95% Confidence Interval** | |
| **Predictor** | **p** | **Odds ratio** | **Lower** | **Upper** |
| Intercept | 0.076 | 0.092 | 0.00659 | 1.285 |
| Gender: | | | | |
| Male–Female | < .001 | 3.8508 | 2.00788 | 7.385 |
| Age groups: | | | | |
| 18–24–55–60 | 0.004 | 54.171 | 3.67342 | 798.848 |
| 25–34–55–60 | 0.953 | 0.9219 | 0.06286 | 13.521 |
| 45–54–55–60 | 0.113 | 10.0052 | 0.57938 | 172.778 |
| 35–44–55–60 | 0.445 | 2.8523 | 0.19407 | 41.921 |
| Educational level: | | | | |
| Bachelor's–High Schooler | 0.001 | 0.1867 | 0.06854 | 0.508 |
| Diploma–High Schooler | 0.984 | 7.56E+07 | 0 | Inf |
| Master–High Schooler | 0.333 | 2.569 | 0.38068 | 17.336 |
| Doctorates–High Schooler | 0.995 | 1.06E+08 | 0 | Inf |

Inf: infinity

Furthermore, our research identified age and educational level as significant factors affecting EMS knowledge. The youngest age group, aged 18 to 24, exhibited the highest levels of EMS knowledge. This reveals that younger generations are more open to receiving information through diverse channels, including social media [14, 15]. Conversely, individuals with a Bachelor's degree demonstrated lower levels of EMS knowledge, which was an unexpected finding. This could be attributed to complacency among highly educated individuals who may assume they already possess sufficient knowledge and, therefore, miss opportunities for further education. To bridge this knowledge gap, it is crucial to develop tailored educational programs targeting both younger age groups and highly educated individuals [16].

Trust in EMS services is also significantly influenced by age and education. Younger participants displayed higher levels of trust due to their greater willingness to embrace change and adapt to new systems. Conversely, those with a Bachelor's degree reported lower trust levels, underscoring the importance of building trust within highly educated segments of the population. Notably, trust in EMS correlated with a willingness to allow paramedics into homes without male escorts, indicating that trust-building efforts can enhance the acceptance of paramedic services, particularly among more conservative communities. Awareness of the unified emergency number 112 was relatively low, with only 76.2% of participants correctly identifying it. This highlights the need for extensive public education campaigns to promote the use of this universal number for all emergencies. A unified emergency number simplifies access to essential services and should complement future EMS awareness initiatives [17].

In terms of expectations, participants generally anticipated EMS response times to be around 30 minutes, which may stem from a lack of understanding regarding EMS operations. This underscores the necessity of educating the public about response time variability based on the nature and location of the emergency [18]. Managing these expectations through educational initiatives can help mitigate potential dissatisfaction with EMS services.

Furthermore, many respondents expressed dissatisfaction with the current EMS service, often rating it as average or poor. To address these concerns effectively, EMS providers should take a proactive approach by improving service quality response times and bolstering public trust. Establishing channels for public feedback and rolling out quality improvement initiatives can play a pivotal role in realizing these objectives [19–21].

While our study aligns with previous research on gender-based differences in EMS awareness and attitudes, such as findings from Aljabri et al. (2012) and Modi et al. (2018), it also introduces some unique insights [6, 8]. We emphasize the potential need for gender-specific approaches in bridging these gaps. Our study's observation that highly educated individuals may exhibit lower EMS knowledge is similar to results from Saberian et al. (2023) and Hamam et al. (2015) and underscores the importance of ongoing education efforts within this demographic [7, 22]. However, we acknowledge that these challenges are not unique to our study and are shared with the research of Alanazi et al. (2012) [23]. Regarding trust, while our findings align with Yonemoto et al. (2018) by indicating higher trust levels among younger participants, it's important to recognize that trust dynamics can vary significantly based on context and generational factors [24].

Our study has several notable strengths that make its findings more reliable and trustworthy. Firstly, we gathered data from a diverse and substantial group of 435 participants in the Eastern region of Saudi Arabia. This extensive sample clearly represents how the public perceives EMS in this area. We also designed a well-structured questionnaire, which underwent a rigorous validation process. This ensured that our collected data accurately reflected people's thoughts and feelings about EMS. Additionally, our study's methodology, including our use of logistic regression analysis, allowed us to thoroughly explore how demographic factors influence people's views on EMS, revealing nuanced relationships. While our study provides valuable insights, it has some limitations. Firstly, we conducted an online survey through social media, which might have led to selection bias. It may have yet to reach people with limited internet access or those who aren't active on social platforms, potentially skewing the sample towards a more tech-savvy and younger group. Additionally, response bias is possible because our data relies on participants' self-reports. People might provide answers they think are socially acceptable or not accurately remember their experiences and feelings.

Moreover, it's important to note that our study concentrated solely on the Eastern region of Saudi Arabia, which limits how broadly we can apply our findings to other regions or countries. Also, because of our cross-sectional design, we can't establish cause-and-effect

relationships between demographic factors and people's perceptions of EMS. Also, it's crucial to acknowledge the potential for response and recall bias inherent in self-reported data. Participants might have provided answers influenced by social desirability or memory lapses, impacting the accuracy of the information gathered. Nevertheless, despite these limitations, our study does offer valuable insights into EMS awareness, knowledge, and trust, which can serve as a solid foundation for future research and targeted interventions.

## 5. Conclusion

In conclusion, our study thoroughly examines how people perceive EMS services in Eastern Saudi Arabia, aligning with our initial goals. By addressing gender and education disparities and factors affecting EMS knowledge and trust, we enhance public engagement with EMS. This involvement can hasten emergency response, emphasizing our key objective of saving lives. Close collaboration among healthcare authorities, educational institutions, and community organizations is vital to translate these insights into action. Utilizing these findings, we can customize educational initiatives, improve emergency protocols, and encourage a proactive emergency culture, ensuring a safer, more responsive community. Future research should explore diverse Saudi regions, employ longitudinal and qualitative methods, and assess awareness campaign effectiveness, enhancing emergency medical services knowledge and trust.

## Supporting information

**S1 File. Author formatting checklist.**
(DOCX)

## Author Contributions

**Data curation:** Abdulkarim Hzazi, Meshal Alharbi.

**Formal analysis:** Abdullah Alruwaili, Saleh Alswaidan, Hassan Alobaid, Ahmed Alomran, Abdulkarim Hzazi, Ibrahim Alhussain.

**Investigation:** Saleh Alswaidan, Hassan Alobaid.

**Methodology:** Ahmed Alanazy, Saleh Alswaidan, Hassan Alobaid, Abdulkarim Hzazi, Ibrahim Alhussain.

**Software:** Ahmed Alomran.

**Supervision:** Ahmed Alanazy, Meshal Alharbi, Meshary Binhotan.

**Writing – original draft:** Ahmed Alanazy, Abdullah Alruwaili, Meshary Binhotan.

**Writing – review & editing:** Ahmed Alanazy, Meshal Alharbi.

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
