## [Decision Letter · Decision Letter 0]

19 Dec 2023

PONE-D-23-35051The Awareness of Public about the Emergency Medical Services and It is Association Factors in the Eastern Region of the Kingdom of Saudi Arabia: A Cross-Sectional StudyPLOS ONE

Dear Dr. Alanazy,

Thank you for submitting your manuscript to PLOS ONE. After careful consideration, we feel that it has merit but does not fully meet PLOS ONE’s publication criteria as it currently stands. Therefore, we invite you to submit a revised version of the manuscript that addresses the points raised during the review process.

We look forward to receiving your revised manuscript.

Kind regards,

Othman A. Alfuqaha, Ph.D.

Academic Editor

PLOS ONE

Journal Requirements:

5. Please amend your list of authors on the manuscript to ensure that each author is linked to an affiliation. Authors’ affiliations should reflect the institution where the work was done (if authors moved subsequently, you can also list the new affiliation stating “current affiliation:….” as necessary).

6. Please include a caption for figure 2.

7. Please include a copy of Table 3 which you refer to in your text on page 8 of 13.

8. We note you have included a table to which you do not refer in the text of your manuscript. Please ensure that you refer to Table 4 in your text; if accepted, production will need this reference to link the reader to the Table.

Additional Editor Comments:

Dear authors,

Thank you for your submission. After careful review, the reviewers have provided valuable feedback on your paper. It is strongly recommended that you carefully consider and address their comments in order to proceed in a positive direction. Additionally, I encourage you to take into account my own comments to further enhance your paper and strengthen its overall quality.

Editor Comments:

Title: Shorten the title to no more than 15 words for conciseness.

Abstract: Include details about your data collection methods such as whether you used random, snowball, or convenience sampling.

Introduction: Clarify whether the deficiency in public awareness of EMS is specific to Eastern Saudi Arabia, the entire Kingdom of Saudi Arabia, or if it extends globally. Broaden the scope in the introduction.

Contributions: Introduce a paragraph outlining the unique contributions of your study and compare them with findings from previous research.

Method Section: Provide more information on how you conducted the inline survey, ensuring it is clear how participants from the Eastern region or other countries were able to respond.

Validity of Scales: Consider adding construct validity measures such as correlations, KMO, Bartlett's test of sphericity, and total variation in addition to content validity and reliability.

Figures: Consolidate figures to present gender and educational levels in a single chart for clarity.

Typos: Carefully review and correct any typos present in the manuscript.

Discussion: Begin the discussion section with a focus on your main findings rather than repeating the study's aims.

International Comparisons: Compare your results with findings from international studies to highlight key differences.

References: Double-check and ensure the accuracy of your references.

We appreciate your commitment to revising your manuscript. Best of luck in this revision journey.

Sincerely,

Dr. Alfuqaha

Reviewers' comments:

Reviewer's Responses to Questions

**Comments to the Author**

1. Is the manuscript technically sound, and do the data support the conclusions?

Reviewer #1: Yes

Reviewer #2: Yes

Reviewer #3: Yes

2. Has the statistical analysis been performed appropriately and rigorously? 

Reviewer #1: I Don't Know

Reviewer #2: Yes

Reviewer #3: Yes

3. Have the authors made all data underlying the findings in their manuscript fully available?

Reviewer #1: Yes

Reviewer #2: Yes

Reviewer #3: Yes

4. Is the manuscript presented in an intelligible fashion and written in standard English?

Reviewer #1: Yes

Reviewer #2: No

Reviewer #3: Yes

5. Review Comments to the Author

Reviewer #1: • Underline number 60 and kindly change from “aged between 18 and 60” to “aged between 18 and 60 years”

• Kindly add exclusion criteria

• How consent was taken for participation in the survey

• Kindly clarify how the survey population were identified for the survey

• How selection bias was minimized in the survey

• Please scan your manuscript again and correct some topological errors

• Please make sure your manuscript was written in line with this journal submission guideline

Reviewer #2: Thank you for inviting me to review this interesting paper which aim to explore the to assess the extent of knowledge regarding EMS within the 53 Eastern Region of Saudi Arabia. The authors did well in obtained the required ethical approval, conducting the study, and reporting the data. In summary, there is need for more awareness public initiatives to increase awareness about EMS. The paper address important gap in the literature and provide a base for staring initiatives in increase the level of awareness in public. However, the manuscript needs further editing to improve readability for the readers.

- Abstract needs editing – for example it would be help full to present the general awareness before presenting the contributing factors such as age, gender, and degree.

- The objective in the introduction presents the study design when authors mentioned “our cross-sectional study …etc.” it would be better to move to the method section.

- Power calculation is fine and adequate try to replace the website link by the software used to calculate the sample size.

- I do not get the grasp from figure 1 for the gender difference.

- Presenting the psychometric properties for the questioner could be confusing. Need more clarification for example how did authors measure reliability (time frame – usually we used 2 weeks intervals between each administration process). I suggest remove the reliability if the questionnaire is valid in measuring what is intended to measure.

- I suggest Knowledge based analysis must be presented first. Then, other factors that can explain the awareness.

- In the logistic regression I would recommend focusing on the adjusted model. Although, the intercepts in tables must be clear in a separate raw.

- Sort the repetition in the manuscript. for example, the aim of the study was introduced gain in the discussion.

Good luck in your revision

Reviewer #3: 1. How the author will make sure that this survey was not done by healthcare staff?

2. How the author will compare between male and female participants about "Emergency Number 112 Awareness" although P value is 0.37 (nonsignificant), Actions when encountering EMS or EMT (P value= 0.62) and Role of a Dispatcher (P value= 0.56)

3. Author mentioned Figures 1 and Figure 2 but there is no "Figure 2"

6. PLOS authors have the option to publish the peer review history of their article (what does this mean?). If published, this will include your full peer review and any attached files.

Reviewer #1: No

Reviewer #2: No

Reviewer #3: No

---

## [Author Response · Author response to Decision Letter 0]

16 Apr 2024

Comment Authors Reply

Journal Requirements:

1) Please ensure that your manuscript meets PLOS ONE's style requirements, including those for file naming. The manuscript was edited according to the provided model

2) Please provide additional details regarding participant consent. In the ethics statement in the Methods and online submission information, please ensure that you have specified (1) whether consent was informed and (2) what type you obtained (for instance, written or verbal, and if verbal, how it was documented and witnessed). If your study included minors, state whether you obtained consent from parents or guardians. If the need for consent was waived by the ethics committee, please include this information. The consent details were added

3) Please include a caption for Figure 2. Caption was added

4) Please include a copy of Table 3 which you refer to in your text on page 8 of 13. Table was added

5) We note you have included a table to which you do not refer in the text of your manuscript. Please ensure that you refer to Table 4 in your text; if accepted, production will need this reference to link the reader to the Table. The table number was edited

Editor Comments:

1) Title: Shorten the title to no more than 15 words for conciseness. The title was shorten 

Abstract: Include details about your data collection methods such as whether you used random, snowball, or convenience sampling. The sampling technique was added to the abstract and in the method section

Introduction: Clarify whether the deficiency in public awareness of EMS is specific to Eastern Saudi Arabia, the entire Kingdom of Saudi Arabia, or if it extends globally. Broaden the scope in the introduction.

Introduce a paragraph outlining the unique contributions of your study and compare them with findings from previous research. The introduction has been updated 

Method Section: Provide more information on how you conducted the inline survey, ensuring it is clear how participants from the Eastern region or other countries were able to respond. Added 

Figures: Consolidate figures to present gender and educational levels in a single chart for clarity.

 our decision to present educational levels with gender distribution aligns directly with our study's objective to assess gender differences in EMS awareness. This approach is crucial to our analysis, providing a nuanced understanding of the interplay between gender and education in the context of EMS knowledge.

Typos: Carefully review and correct any typos present in the manuscript.

 The manuscript was checked

Discussion: Begin the discussion section with a focus on your main findings rather than repeating the study's aims.

International Comparisons: Compare your results with findings from international studies to highlight key differences.

 the dissection has been updated 

References: Double-check and ensure the accuracy of your references. Checked 

Reviewer #1 comments

1) Underline number 60 and kindly change from “aged between 18 and 60” to “aged between 18 and 60 years The word “years” was added

2) Kindly add exclusion criteria The exclusion criteria were added

3) How consent was taken for participation in the survey The consent details were added

4) Kindly clarify how the survey population were identified for the survey

5) How selection bias was minimized in the survey Details regarding the survey population identification were added

6) Please scan your manuscript again and correct some topological errors 

7) Please make sure your manuscript was written in line with this journal submission guideline The manuscript was edited according to PLOS ONE criteria

Reviewer #2 Comments

1) Abstract needs editing – for example it would be help full to present the general awareness before presenting the contributing factors such as age, gender, and degree. The awareness was introduced first.

2) The objective in the introduction presents the study design when authors mentioned “our cross-sectional study …etc.” it would be better to move to the method section. This sentence was edited accordingly.

3) Power calculation is fine and adequate try to replace the website link by the software used to calculate the sample size. This site has no related software

4) I do not get the grasp from figure 1 for the gender difference. The figure number was edited

5) Presenting the psychometric properties for the questioner could be confusing. Need more clarification for example how did authors measure reliability (time frame – usually we used 2 weeks intervals between each administration process). I suggest remove the reliability if the questionnaire is valid in measuring what is intended to measure. We conducted the reliability test to enforce our study.

6) I suggest Knowledge based analysis must be presented first. Then, other factors that can explain the awareness. The awareness was introduced first.

7) In the logistic regression I would recommend focusing on the adjusted model. Although, the intercepts in tables must be clear in a separate raw. The we prefer to keep both models to provide a comprehensive overview. The intercepts in tables are already in separate cells.

8) Sort the repetition in the manuscript. for example, the aim of the study was introduced gain in the discussion. The manuscript was reviewed and repetition was removed.

Reviewer #3 Comments

2) How the author will compare between male and female participants about "Emergency Number 112 Awareness" although P value is 0.37 (nonsignificant), Actions when encountering EMS or EMT (P value= 0.62) and Role of a Dispatcher (P value= 0.56) While the p-values were non-significant, our study's findings contribute to the broader understanding of emergency response awareness and behavior, suggesting the need for further exploration of gender-related factors.

3) Author mentioned Figures 1 and Figure 2 but there is no "Figure 2" The figure number was edited

---

## [Editor Report · Decision Letter 1]

26 Jun 2024

The Awareness of Public about the Emergency Medical Services in the Eastern Region of Saudi Arabia

PONE-D-23-35051R1

Dear Dr.

We’re pleased to inform you that your manuscript has been judged scientifically suitable for publication and will be formally accepted for publication once it meets all outstanding technical requirements.

Kind regards,

Othman A. Alfuqaha, Ph.D.

Academic Editor

PLOS ONE

Additional Editor Comments (optional):

Congratulations on your diligent efforts.
---

## [Editor Report · Acceptance letter]

2 Jul 2024

PONE-D-23-35051R1 

PLOS ONE

Dear Dr. Alanazy, 

I'm pleased to inform you that your manuscript has been deemed suitable for publication in PLOS ONE. Congratulations! Your manuscript is now being handed over to our production team.

Kind regards, 

on behalf of

Dr. Othman A. Alfuqaha 

Academic Editor

PLOS ONE